

# Biomass burning smoke transport and radiative impact over the city of Sao Paulo: An extreme event case study

Jorge Rosas Santana[1], Gabriela Lima da Silva [1], Marcia Akemi Yamasoe[1,] Nilton Èvora do Rosario[2]

[1] Departamento de Ciências Atmosféricas, Instituto de Astronomia, Geofísica e Ciências Atmosféricas, Universidade de São Paulo (IAG-USP), São Paulo, 05508-090, Brazil.
[2] Departamento de Ciências Ambientais, Instituto de Ciências Ambientais, Químicas e Farmacêuticas, Universidade Federal de São Paulo, São Paulo, 09972-270, Brazil.

*Correspondence to*: Jorge Rosas Santana(jrosas43@email.com)

**Abstract.** Biomass burning is a worldwide practice applied to deforestation which can have disastrous consequences to local and regional environments. This paper describes a case study of an extreme event of biomass burning smoke transport toward the São Paulo metropolitan area (MASP), on 19 August 2019, when the city experienced an uncommon completely dark sky around 3:00 PM. A synergy between air mass back trajectories, satellite retrieved aerosol fields and surface radiometric measurements was used to find the origin of the smoke plume affecting the city and to analyse the radiative impact of the transport of the smoke toward the city. Results showed that the MASP atmosphere was affected by the transport of a dense smoke plume with aerosol optical depth at 550 nm above 1. Air mass back trajectories and auxiliary data indicated that most of the smoke was emitted two days before arrival. The smoke plume in combination with clouds, associated with a frontal system, produced a strong radiative impact, as observed by a regional network of pyranometers. During the darkness day, the diurnal clearness index was below 0.1 in all five MASP stations and a maximum of the cloud optical depth higher than 300 was retrieved producing irradiances at surface dropped to 0 during approximately 40 minutes. The strong radiative efficiency (cloud radiative effect per cloud optical depth unit) of this extreme event, was 7% higher than other overcast days observed in a two-year period.

## 1 Introduction

Biomass burning (BB) is one of the most important sources of aerosol particles in the tropical regions driven globally by activities such as deforestation, clearing residual fields, pest control and grassland management. The smoke particles influence air quality with several health consequences (Keywood et al., 2015), mainly related to respiratory diseases. Some aerosol particles, such as the polycyclic aromatic hydrocarbons, have carcinogenic properties (Pereira et al., 2017). Significant reduction in visibility under smoke scenarios have been also reported (Lee et al., 2017), which represents a problem for the transport sector, by lowering the visibility, for instance can cause flight delays and road accidents.

Aerosols from BB have significant radiative impacts in the atmosphere, directly and indirectly, affecting also its dynamics and the circulation of air masses. Radiative effects of BB can lead to a reduction of sensible and latent heat fluxes near the surface, the boundary layer height and wind speed and to an increase in relative humidity (Li et al., 2022). Above the south-eastern





Atlantic Ocean, BB aerosols were found to influence the vertical circulation, by warming the layers where they are present (Mallet et al., 2020). Chang et al. (2021) and Mallet et al. (2020) showed stronger radiative cooling at the surface during BB events in Australia comparable to volcanic eruptions. They reported a reduction of temperature between 3.7 to 4.4 °C, with a

radiative effect around -14.8 to -17.7 W m$^{-2}$. In addition, in the presence of BB aerosols, during the dry season, there is an increase of the gross primary productivity in the Amazon Forest (Moreira et al., 2017). This is due to the increase in photosynthetic activity caused by the enhancement of diffuse radiation, which results from the scattering of solar radiation by these aerosols (Moreira et al., 2017; Yamasoe et al., 2006).

In the well-known "Twomey effect" (Twomey, 1977), given the same amount of water vapor, the cloud albedo is expected to

be enhanced due to the increase of atmospheric aerosol concentration. However, the response of clouds to the aerosol particles depends on the characteristics and the dynamics of the cloud systems. Koren et al. (2004) observed a reduction in the cumulus cloud cover in the presence of heavy smoke in the Amazon region. By contrast, Brioude et al. (2009) showed an increase in maritime cloud fraction and cloud albedo associated with a biomass burning aerosol layer above the marine boundary layer. For strong convective clouds, the effect of "cloud invigoration" due to the increase of aerosol concentration leads to a chain

of events strengthening the upwards currents inside clouds (Altaratz et al., 2014; Rosenfeld et al., 2008). Gautam et al. (2016b) found strong impacts in the spectral radiance values measured by satellite when clouds are embedded in a smoke layer.

In South America, BB aerosols emitted by vegetation fires in the central and northern portions of the continent are often transported towards the west and south of the continent. When cold front systems from the south penetrate the continental area, polluted air masses from these biomass burning regions can be pushed south-eastward reaching the most populated urban areas

of Brazil (Freitas et al., 2005). Those polluted events are more frequent from August to October (Landulfo et al., 2003), leading to significant changes in the regional aerosol optical properties and radiative impacts (Rosário et al., 2013; Yamasoe et al., 2017). Eventually, under certain circumstances, smoke transported also affects particulate matter concentrations at the surface causing a degradation of air quality (Castanho et al., 2008; de Miranda et al., 2017). In Brazil, biomass burning activity presented a declining trend in the first decade of the 21st century, but an increasing trend has been observed in recent years,

particularly over the Amazon rainforest, Pantanal and Cerrado biomes (Mataveli et al., 2021; Rosário et al., 2022).

In August 2019, large areas in the central portion of South America, particularly in the deforestation arch of the Brazilian Amazon, underwent an outbreak of forest fires that had strong consequences for the flora, fauna, and nearby human settlements. There was a high increase (150%) in the seasonal mean aerosol optical depth at 550 nm compared to 2018 (from 0.15 to peaks of 0.4) in South America. As a response, a 30% increase in the cooling effect at the surface in the Amazon wildfire region was

observed (with radiative forcing values from -29 W/m-2 to -38 W/m-2) (Yuan et al., 2022). Bencherif et al. (2020) identified two major successive events of transportation of biomass burning aerosols from the Amazon to southern South America during August 2019. In the second event, starting on August 14th, a massive aerosol plume moved toward south-eastward direction, reaching the Metropolitan Area of São Paulo (MASP) by August 19th and, in combination with the cloudiness, contributed to a sudden darkness in broad daylight, and later on that day, dark rain was observed, what has been called a "black" rain

event  (Pereira et al., 2021).



Focused on this event of August 19, this paper analyses the transportation of smoke and the impact on the downward shortwave irradiance at the surface integrating modelling and observational dataset. The observational dataset includes a synergy between a regional surface broadband irradiance measurement network, aerosol optical properties, backscatter profiles which were also employed to characterize the source of the plume arriving at the MASP.

This paper is structured as follows: the materials and methods section describes the study area; the dataset used in the analysis and provides details on the synergies and the retrieval methods, and the modelling systems used. In the results section, we analyse the source and spatial distribution of the smoke plume transported toward MASP and discuss the radiative impact of this transport event on the regional pyranometer network. More detailed analysis is presented for the Metropolitan Area of São Paulo (MASP), including changes in cloud optical depth. The conclusions are wrapped up with the main findings and

suggestions for further studies.

## 2 Materials and methods

### 2.1 Study region and ground-based measurements

Table 1 lists the instruments, the temporal resolution of measurements, the network names or locations of each radiometer and the variables employed from each one. We analysed global horizontal irradiance (GHI) measurements from 18 pyranometers

operated by the Instituto Nacional de Meteorologia (INMET). Additionally, GHI measurements were obtained from two instruments maintained by the Laboratório de Radiação e Aerossóis Atmosféricos (LRAA) from the Instituto de Astronomia, Geofísica e Ciências Atmosféricas (IAG): CNR4 Net Radiometer located at the Parque de Ciência e Tecnologia  of the University of São Paulo (CIENTEC) and a pyranometer located in the roof of the Pelletron building at the Instituto de Física da Universidade de São Paulo (IFUSP). From the GHI, we computed the diurnal clearness index ($K_t$), defined as the ratio of

diurnal GHI at the surface to the equivalent value at the top of the atmosphere (TOA). Furthermore, the normalized cloud radiative effect (NCRE) is estimated using GHI along with radiative transfer simulations (Section 1.4).

We obtained data of version 2.0 of column integrated precipitable water vapor (PWV), aerosol optical depth at 500 nm ($AOD_{500}$) and Angström coefficient estimated from AOD at the pair of 440 nm and 870 nm ($\alpha$) from measurements of AERONET (Aerosol Robotic Network) (Holben et al., 1998) sun-photometer installed at IF, version 2.0.

**Table 1 - Radiometer type, temporal resolution of data acquisition, the institution responsible for the specific network or location of each radiometer and analysed/estimated variables. In parentheses are the number of sites analysed in this study or the variable unit.**

| Radiometer | temporal resolution | network/location (number of sites) | Variables (unit) |
|---|---|---|---|
| Pyranometer | 1-hour | INMET (18) | GHI (W m$^{-2}$), $K_t$ |
| Pyranometer | 1-minute | IFUSP - LRAA (1) | GHI (W m$^{-2}$), |





| CNR4 | 1-minute | CIENTEC (1) | GHI (Wm$^{-2}$), COD, NCRE |
| sun-photometer | 15-minutes | AERONET (1) | AOD$_{500}$, PWV (cm), α |

Figure 1 depicts the region of study, which includes part of the central, southeast and south portions of Brazil, and the locations of the surface stations. The stations are grouped into 4 sub-areas: stations located in the Metropolitan Area of São Paulo
(eastern part of São Paulo state) (MASP); stations located in Mato Grosso do Sul state (MS); with stations located in Paraná state (PR); stations located at the northwestern portion of São Paulo state (SP-W). Table 2 presents a geographical summary of each station and the acronyms that will be used throughout the paper.

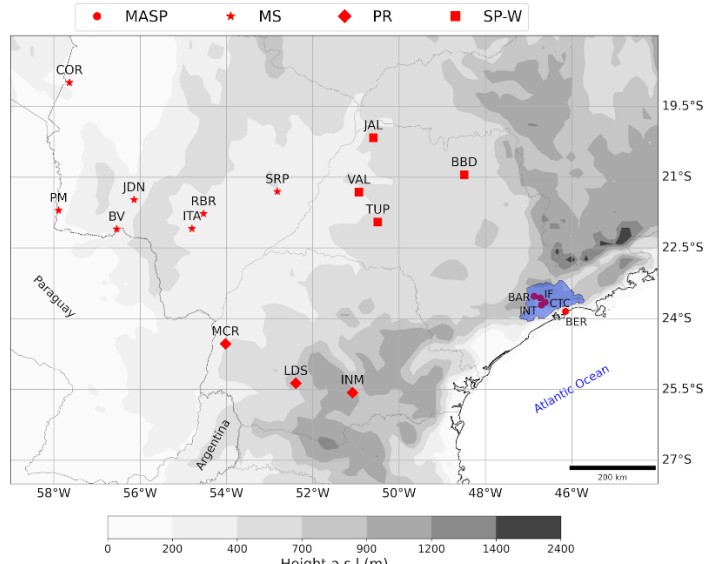

**Figure 1 - The regions of Brazil analyzed in this study: Central (MS), Southeast (SP) and South (PR). The red symbols indicate the**
**locations of the meteorological stations considered in the analysis and their different formats indicate the related administrative domain (SP-W - western portion of Sao Paulo state; MS - Mato Grosso do Sul state; PR - Paraná state; MASP - Metropolitan Area of São Paulo). The blue area highlights the MASP area.**

**Table 2 - Location of INMET stations and two other data sources (CIENTEC and IFUSP) analysed in this study. * hasl - height above sea level.**

| station | lat (°) | lon (°) | region | acronym | hasl* (m) |
|---|---|---|---|---|---|
| Porto Murtinho | -21.70 | -57.88 | MS | PM | 79 |
| Corumbá | -18.99 | -57.64 | MS | COR | 112 |
| Jardim | -21.48 | -56.14 | MS | JDN | 252 |
| Bela Vista | -22.10 | -56.54 | MS | BV | 213 |
| Itaporã | -22.09 | -54.80 | MS | ITA | 386 |
| Rio Brilhante | -21.77 | -54.53 | MS | RBR | 324 |



| Santa Rita do Rio Pardo | -21.31 | -52.82 | MS | SRP | 394 |
|---|---|---|---|---|---|
| Inácio Martins | -25.57 | -51.08 | PR | INM | 1159 |
| Laranjeiras Do Sul | -25.37 | -52.39 | PR | LDS | 861 |
| Marechal Cândido Rondon | -24.53 | -54.02 | PR | MCR | 386 |
| Valparaíso | -21.32 | -50.93 | SP-W | VAL | 385 |
| Tupã | -21.95 | -50.49 | SP-W | TUP | 489 |
| Jales | -20.17 | -50.59 | SP-W | JAL | 447 |
| Bebedouro | -20.95 | -48.49 | SP-W | BBD | 572 |
| Barueri | -23.52 | -46.87 | MASP | BAR | 756 |
| Bertioga | -23.84 | -46.14 | MASP | BER | 19 |
| Interlagos | -23.71 | -46.70 | MASP | INT | 758 |
| CIENTEC | -23.65 | -46.62 | MASP | CTC | 792 |
| IFUSP | -23.56 | -46.73 | MASP | IF | 748 |


## 2.2 Satellite-derived and reanalysis products

Table 3 describes the products of satellites employed, as well as reanalysis data. The remote sensing products from satellites analysed were sampled from areas surrounding each ground station. For each station, measurements within a radius of 0.4° were averaged, with the weights assigned based on proximity to the station to ensure a more accurate representation of local

conditions. In the following, a brief description of each analysed product is presented.

Instantaneous all sky radiative fluxes at the top of the atmosphere and at the surface (GHI) were obtained from the Clouds and the Earth's Radiant Energy System (CERES) project (Loeb et al., 2016; Scott et al., 2022). The data from CERES used the sensor Single Scanner Footprint (SSF), with horizontal resolution of 20 km at nadir. The data was provided by the Langley Research Center (LARC) of the National Aeronautics and Space Administration (NASA) at the website

https://ceres.larc.nasa.gov/data/.

**Table 3: Description of satellite and reanalysis data used, including the spatial resolution, the source and variables employed.**

| Product | Resolution | Source | variables |
|---|---|---|---|
| CERES SSF/CRS-level2 | 20 km at nadir | Terra/Aqua satellite | irradiances at TOA and GHI at the surface |
| MODIS: MYD04_3K, MOD04_3K | 3 km | Aqua and Terra satellites | $AOD_{550}$ |
| MOD14 and MYD14 | 1 km | Aqua and Terra satellites | Firespot data |



| M2T1NXAER | 0.625° x 0.5° | MERRA2 reanalysis | AOD$_{550}$ for overcast conditions |
|---|---|---|---|
| CAL_LID_L1-Standard-V4-10 | 1 km along track | CALIPSO satellite | Profiles of attenuated backscatter at 532 nm |
| OR_ABI-L2-MCMIPF-M6_G16 | 2 km | GOES-16 | Data from multiple spectral bands of the ABI (Advanced Baseline Imager), specifically channels 2 (0.64 μm), 3 (0.86 μm), and 1 (0.47 μm). |
| ERA5 | 25 km | ERA-5 | Pressure at height above sea level (HASL) and wind at 850 hPa |

Moderate Resolution Imaging Spectroradiometer (MODIS) sensors are aboard the polar orbit satellites Aqua and Terra, giving a good horizontal resolution, but with the limitation of delivering information at specific local times for each point overpassed.

Aerosol optical depth at 550 nm (AOD550) from the MODIS level 2 products using the dark target methodology (Levy et al., 2013) and with 3 km by 3 km horizontal resolution were obtained from https://ladsweb.modaps.eosdis.nasa.gov/ (last access: July 2024).

In addition, vertical profile information of aerosol was taken from the polar-orbit Cloud-Aerosol Lidar and Infrared Pathfinder Satellite Observation (CALIPSO) (Winker et al., 2007). CALIPSO performs the vertical scanning of the

atmosphere, ensuring a better detection of the aerosol layers. We analysed the total attenuated backscatter signal at 532 nm and the aerosol classification from the Lidar Level 1B profile data, version 4-10 product of CALIPSO (NASA/LARC/SD/ASDC, 2016) to help identify the smoke layers.

The Fire Radiative Power (FRP) data from MODIS (MOD14 and MYD14) (Giglio et al., 2016) was used to determine the wildfire spots in Brazil during the study period which potentially contributed to the smoke transport event over southeast

of Brazil. FRP data was provided by NASA's Fire Information for Resource Management System (FIRMS). Only measurements with a confidence level higher than 50 (on a scale from 0 to 100, with 0 being the lowest quality and 100 the highest) were selected for use in this study.

GOES-16 data from the Advanced Baseline Imager (ABI) sensor provides detailed information on cloud cover and the presence of aerosol plumes, with a horizontal resolution of 2 km. Specifically, we use a combination of channels 2 (0.64 μm), 3 (0.86

μm), and 1 (0.47 μm) to depict the presence of clouds and aerosols.

We employed ancillary information from ERA-5, including pressure fields at sea level and wind at 850 hPa. Pressure fields help in identifying synoptic-scale systems Additionally, wind data at 850 hPa provides information on low-level wind flows that play a role in the transport of aerosol plumes.

When the scene was cloudy, aerosol products were not retrieved from any satellite or ground-based measurements.

In those cases, we needed to estimate the aerosol parameters equivalent to clear skies, in order to evaluate the radiative impact due only to clouds. AOD hourly mean values for cloudy conditions were obtained from the M2T1NXAER Modern-Era Retrospective analysis for Research and Applications version 2 (MERRA-2) (Randles et al., 2017).



A bias correction of the AOD at 550 nm from MERRA2 was conducted by comparing it with colocated AOD at 550 nm from AERONET 2008–2019 for the months of July, August, and September. Since AERONET does not provide AOD at this wavelength directly, the estimation was made using α. The bias was corrected using simple linear regression of data spanning the period. Error propagation (not shown) resulted in a final uncertainty of 0.06 for the bias-corrected AOD at 550 nm.

### 2.3 HYSPLIT and the sources of the aerosol smoke plume

The Hybrid Single-Particle Lagrangian Integrated Trajectory (HYSPLIT) (Draxler and Hess, 1998) is a dispersion model which allows the estimation of the air mass trajectories with Lagrangian and Eulerian approaches. A variety of simulation methods can be applied using different meteorological model outputs. For this study we used HYSPLIT to compute backward trajectories of air masses that arrived in São Paulo on August 19th, using as input the meteorological data from the GFS (Global Forecast System) with 0.25° horizontal resolution.

The HYSPLIT model simulated the route of air parcels that arrived over the CTC site at different times (from 03:00 UTC on the 19th to 02:00 UTC on the 20th, every hour) and selected altitudes above ground level (2, 3 and 4 km). The backward position of each trajectory was calculated until 06:00 UTC on August 17th, 2019, to coincide with CALIPSO overpass time. A total of 72 air parcel trajectories were simulated. Using the HYSPLIT model, a heat map was created to show the origins of air parcels arriving at CIENTEC throughout August 19, based on their positions at 06:00 UTC on August 17. This analysis allowed us to estimate the regions from which these air parcels originated and to verify whether these areas correspond to the wildfire locations and smoke layer observed in the CALIPSO measurements.

### 2.4 Downward shortwave irradiance simulations and cloud optical depth retrievals

Broadband simulations of GHI at the surface for clear-sky and cloudy conditions at the CTC station were performed using the radiative transfer model LibRadtran (Emde et al., 2016; Mayer and Kylling, 2005). Simulations were carried out for overcast days, including the extreme smoke event of August 19th to retrieve the cloud optical depth (COD) and the NCRE. For each measurement timestamp on every overcast day, aerosols, cloud cover, surface conditions, and water vapor were retrieved or inferred and used as inputs for GHI simulations.

We used $AOD_{550}$ with bias correction, and α from MERRA2. PWV data from AERONET was interpolated for the overcast days using PWV retrieved before and after each day. We assumed a spectrally homogeneous surface albedo of 0.15, typical of urban regions at midlatitudes (Sailor and Fan, 2002). The solar spectral irradiance at TOA from Gueymard (2004) was adopted. Cloud properties, assumed as a homogeneous plane-parallel layer, were also included with fixed values for the effective radius of 11.6 μm, cloud base and top of 893 m and 1700 m, respectively, obtained from CLOUDSAT (Austin, 2007) climatology for the MASP region. Simulations were performed at every 1 minute following the frequency of surface irradiance measurements at the CTC, in the MASP.

To obtain the COD, the cloud modification factor (CMF) was simulated by LibRadtran for completely overcast sky conditions. CMF is defined as the ratio between the measured GHI at the surface to the GHI estimated for clear sky conditions. COD was estimated by iteratively adjusting COD values and computing the CMF, until the difference between the simulated and





measured CMF converged to a very low value, considered as less than 0.05%. CMF was used for retrieving COD because it is strongly associated with transmittances of clouds (Serrano et al., 2015). Propagation error of the retrieval (not shown) shows
an error of the retrieval of 18 % considering employing a CNR4. When comparing the retrieval of COD using CMF with the one estimated with Multifilter Rotating Shadowband Radiometer (MFRSR) at 415 nm measurements for overcast days (Leontyeva and Stamnes, 1994) we observed a maximum difference of 15%.

The solar broadband cloud radiative effect at the surface was estimated using NCRE, obtained as presented in Eq. (1). The superscripts clr and obs refer, respectively, to the surface irradiances simulated for clear sky using LibRadtran and observed
under cloudy conditions from CNR4.

$$NCRE = \frac{GHI^{obs} - GHI^{clr}}{GHI^{clr}} \qquad (1)$$

## 3 Results

### 3.1 MASP's air parcels origin on August 19

Figure 2 depicts $AOD_{550}$ map from MODIS-Aqua on August 17, 2019, between 17:00 to 18:00 UTC and the profile of total
attenuation backscatter signal from CALIPSO at 532 nm on the same day, between 05:42 and 05:49 UTC. Note, over South America, between 14º to 18° S and 61° to 62° W, a layer with the bottom at about 1.5 km and the top at 3 km, whose total attenuated backscatter values varied from 1.5 m-1s-1 to 5 m-1sr-1. This layer was identified as subtype polluted continental/smoke with elevated smoke by the CALIPSO retrieval algorithm.



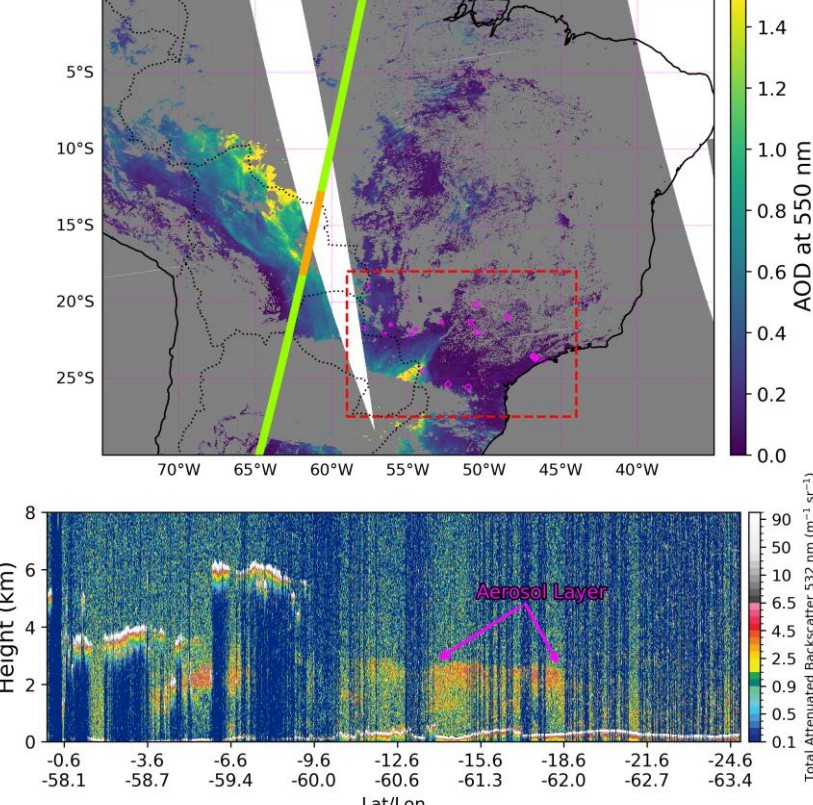

**Figure 2: MODIS aerosol optical depth (AOD) at 550 nm (top) for the Aqua overpass on August 17, 2019, between 15:10 UTC and 20:05 UTC, and the CALIPSO vertical profile of total attenuated backscatter at 532 nm during the overpass over South America on August 17, 2019, from 05:42 to 05:49 UTC (bottom). In the top panel, the rectangle highlights the study area, and the stations as in Fig 1. The line represents the CALIPSO overpass (width not to scale), with the aerosol layer region highlighted. In the bottom panel, arrows indicate the position of the aerosol smoke plume, as classified by the CALIPSO algorithm.**

Figure 3 shows the area density of the origin of air parcels at 6:00 UTC on August 17th, which arrived at MASP on August 19, the fire spots and the CALIPSO satellite overpass track on August 17th. The trajectories of air mass originated around the border of Brazil and Bolivia, where a high number of fire spots were detected. Note that the aerosol layer observed by the CALIPSO overpass crossed this region characterized by high density of air parcels and high fire spot number.

Figure 4 shows the HYSPLIT back trajectories arriving at MASP (CTC station) at three altitude levels: 2, 3, and 4 km, at 6:00, 12:00, 18:00, and 21:00 UTC. The air parcels followed a northwest to southeast trajectory, originating in the Brazilian middle-west, east of Bolivia and north of Paraguay. The parcels' altitude increased with time, except for the ones arriving at MASP at 12:00 UTC and at the altitude of 2 km above ground. Some parcels originated near the surface two days before, especially those that arrived at CTC at 3 km at 12:00 UTC and 21:00 UTC and parcels arriving at 2 km around 18:00 UTC.



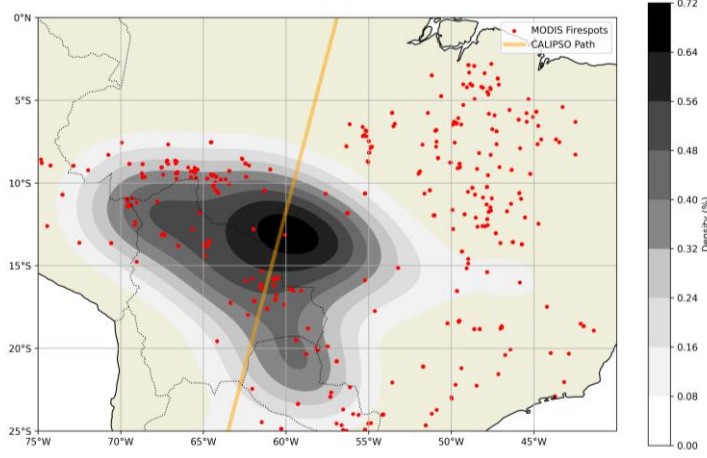


**Figure 3: Density of air parcel's departure position (shaded) toward CTC estimated with HYSPLIT back trajectories analysis at 06:00 UTC on August 17th for arrival heights of 2, 3 and 4 km, the CALIPSO's path on that day (width not in scale) and wildfire counts (red dots) are also plotted.**

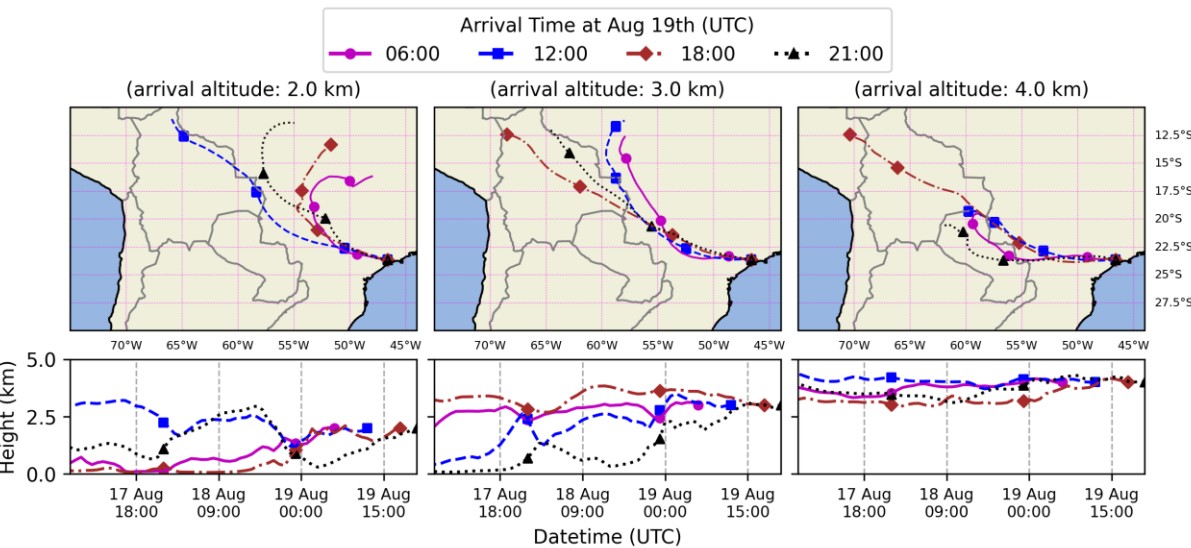

**Figure 4: Air parcels which arrived at CTC in MASP, at 06:00 UTC, 12:00 UTC, 18:00 UTC and 21:00 UTC on August 19th, 2019, at three altitude levels (2, 3 and 4 km). Markers represent the last hour of each day.**

## 3.2 Regional radiative impact of the smoke

This section analyses the transport of the smoke plume and what happened to the surface solar radiation along the way and when/where the smoke and the clouds mixed, using satellite-derived products, reanalysis data, and surface measurements

collected at the meteorological stations network.



Figure 5 presents GOES-16 natural colour RGB imagery combined with 850 hPa wind vectors and pressure field at sea level, covering the study region from August 16 to 19 at 16:00 UTC. Clear skies were predominant on August 16 and 17, with stronger northerly winds observed in the western part of the region. On August 18, there was a noticeable increase in smoke, accompanied by westerly winds, and clouds extending across Paraná (MS), parts of Mato Grosso do Sul (MS), and São Paulo (SP-W). A cold front reached the Metropolitan Area of São Paulo (MASP) on August 19, pushing the smoke plume observed above the cloud layers toward MASP. Consequently, the combination of clouds and smoke became more prevalent between August 18 and 19 across the southeast region, and especially over MASP on August 19.

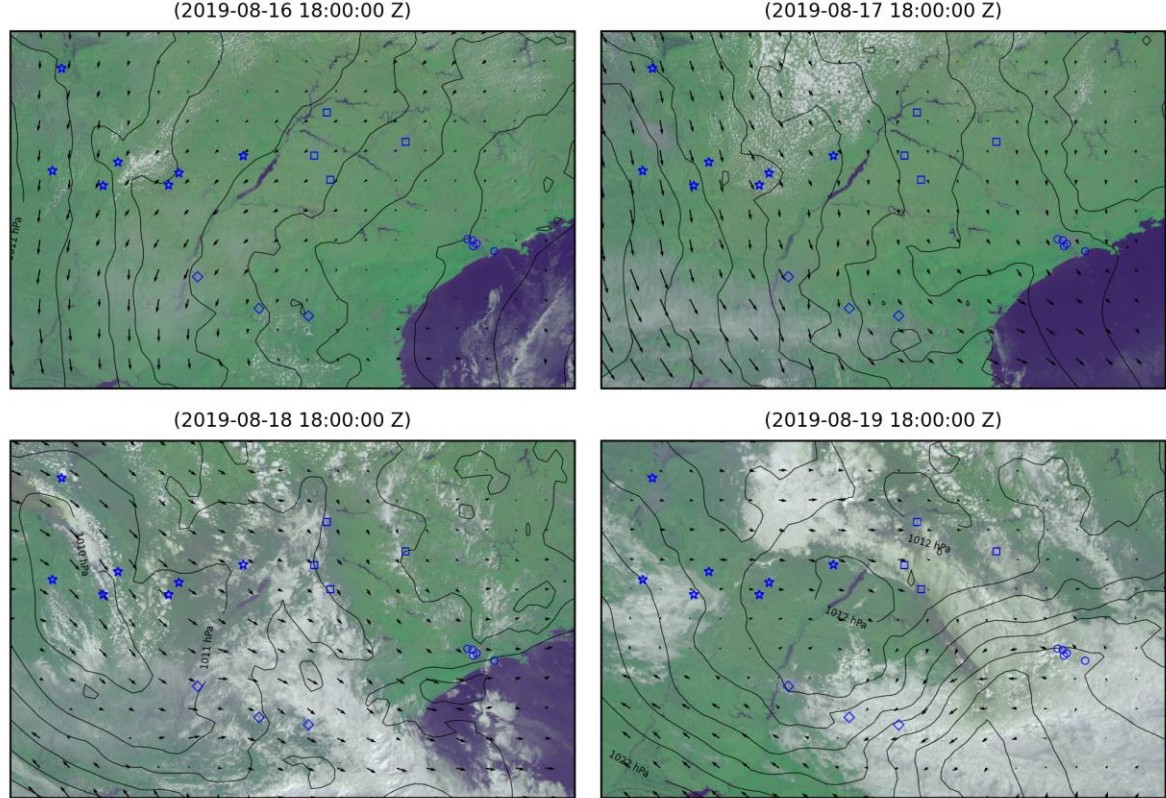

**Figure 5: Sequence of RGB satellite maps in natural colour using channels 2 (0.64 µm), 3 (0.86 µm), and 1 (0.47 µm) from GOES-16, allowing observation of cloud cover and the smoke plume. The maps also display wind components at 850 hPa (arrows) and mean sea level pressure isobars (lines) from ERA5 Reanalysis around 18:00 UTC (15:00 LT), from August 16 to August 19. The selected region is the same as in Fig. 1 and includes the stations, shown as markers.**

Daily time series of the maximum AOD at 550 nm from MODIS and MERRA-2 are shown in Figure 6, with MERRA-2 AOD included only when the corresponding MODIS data were unavailable within a maximum distance of 0.4°. AOD increase was first observed at stations near the wildfires' source (MS), followed by PR state, around August 18, and one day later at stations in the eastward regions (SP-W and MASP). Note that maximum AOD values, exceeding 1.1, at MASP were estimated by MERRA-2 with bias correction on August 18, 19 and 20. A particularly significant event occurred on August 19 at MASP, marked by strong darkness due to the combination of smoke and clouds.



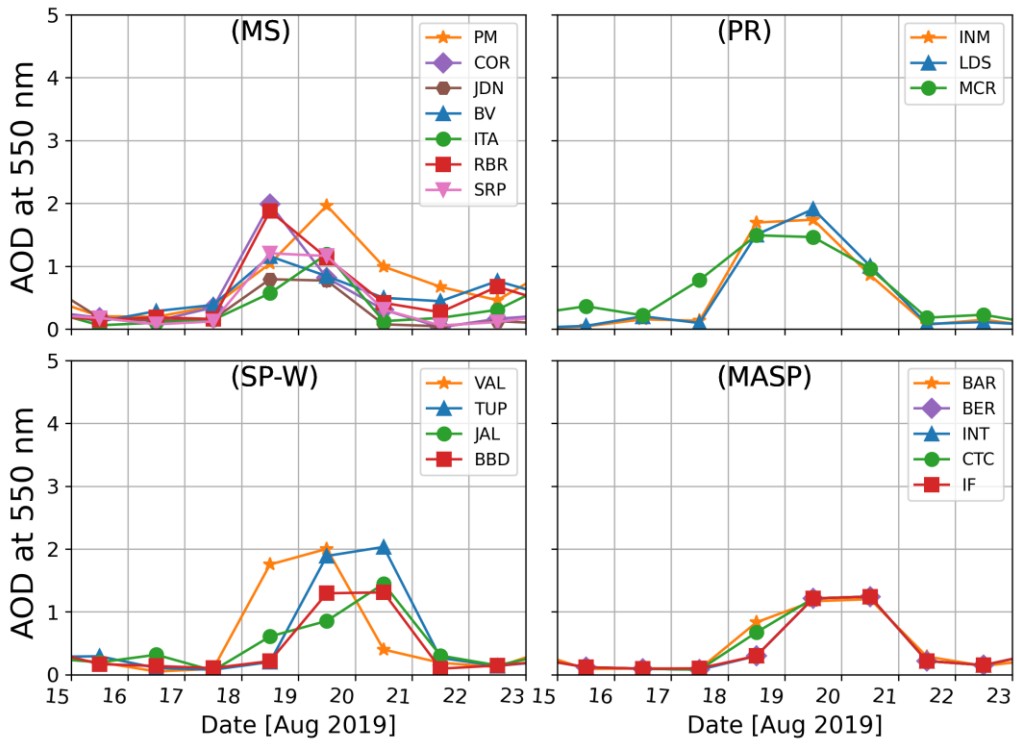

**Figure 6. Time series of daily maximum AOD at 550 nm measured by MODIS or estimated by MERRA 2 over each meteorological station considered in this study (INMET, IF and CTC). MERRA-2 AOD was included only when the corresponding MODIS data were unavailable within a maximum distance of 0.4°.**

Figure 7 shows all-sky values of upward irradiance at TOA at 13:00 and 18:00 UTC from CERES on board Terra and Aqua satellites, respectively. On August 18, irradiance increased over the MS region, coinciding with higher AOD at 550 nm, and exceeded 250 W m⁻² over station SRP, in the MS state. In other regions, the increase of irradiance was attributed to the combined effects of clouds and smoke. The SP-W region showed increased upward shortwave irradiance at the TOA between August 18 and 19. The highest TOA irradiance was observed over MASP on August 19, reaching above 500 W m⁻², an increase of 350 W m⁻² compared to previous days.





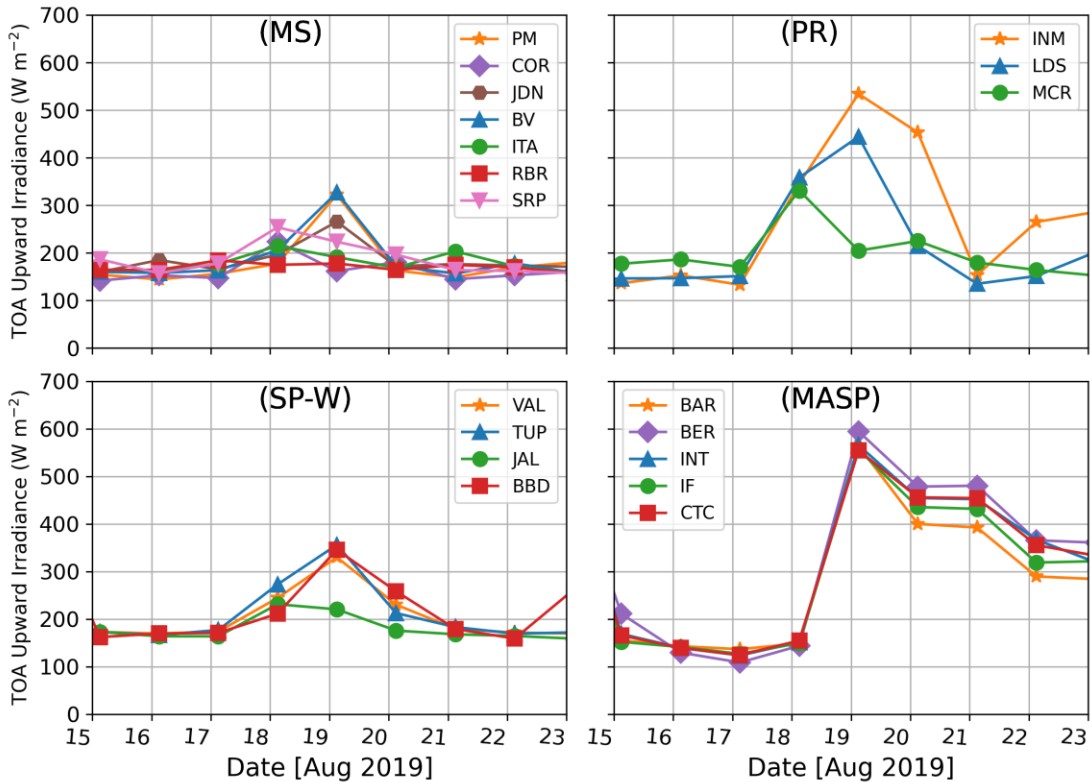


**Figure 7: Upward solar irradiance at the top of the atmosphere (TOA) estimated from CERES at the time of the overpasses at each region at 13:00 and 18:00 UTC.**

Figures 8 and 9 present the evolution of the all-sky daily mean of GHI at the surface estimated from CERES, along with the

$K_t$ from ground-based measurements. Note that the CERES mean values only take into account hours between 13 and 18 UTC while the ground-based data consider hours from sunrise to sunset. As previously noted, an increase in smoke combined with clouds is observed starting on August 18, particularly in the MS region in the northeast of the study area and the PR region in the southwest, influenced by the cold front approaching. The MS stations showed a minimum in irradiance one day before the smoke-affected day in MASP, with $K_t$ values near 0.4 and CERES downward irradiance around 400 W m⁻². In addition, the

southernmost PR region experienced lower values due to stronger cloud front influence, with $K_t$ near 0.3 and average GHI from CERES around 200 W m⁻².

In the MASP, on August 19, all stations agree with daily GHI dropped from 600 W m⁻² on the previous day to around 100 W m⁻² (Fig. 8), while $K_t$ decreased from 0.6 to near 0.1 (Fig. 9). The daily $K_t$ values near of 0.1 in MASP falls below the 1st percentile of the distribution from the past two years of all-sky observations in the region, indicating an extreme reduction in

surface downward shortwave irradiance.



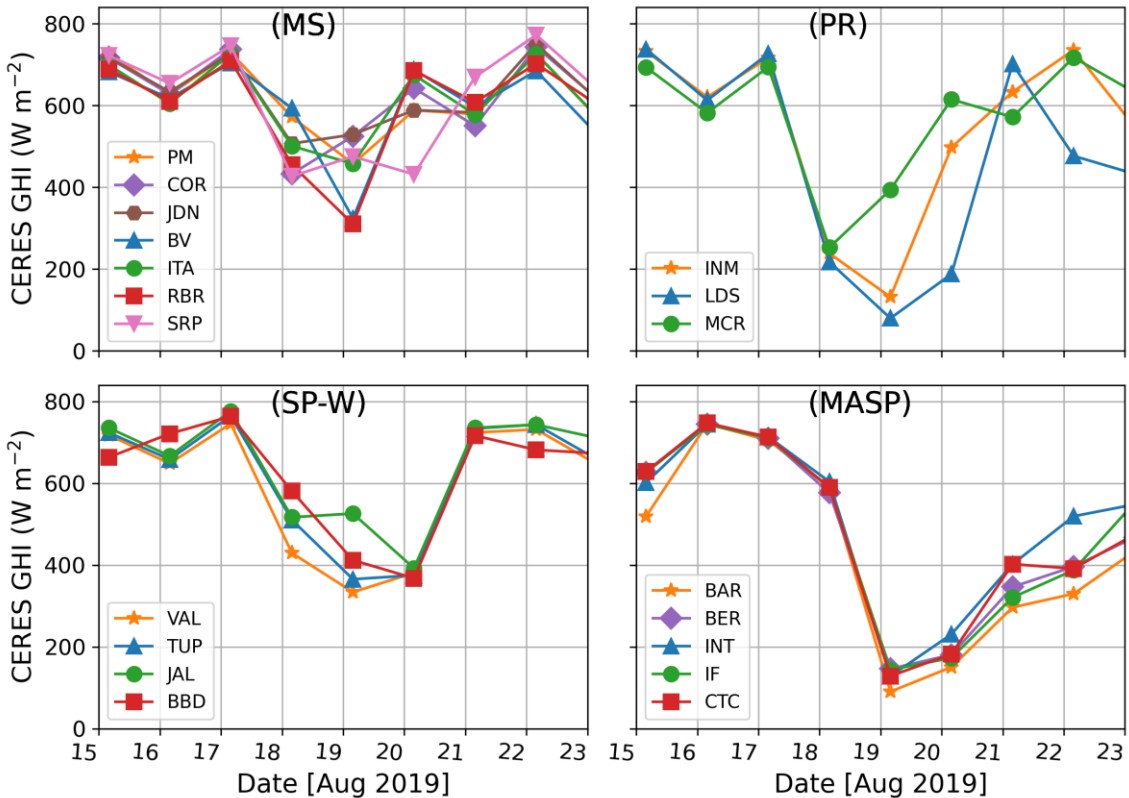

**Figure 8:**
CERES daily mean all sky GHI estimated at INMET, IF and CTC stations. Daily mean values were computed only using the available hours from CERES.






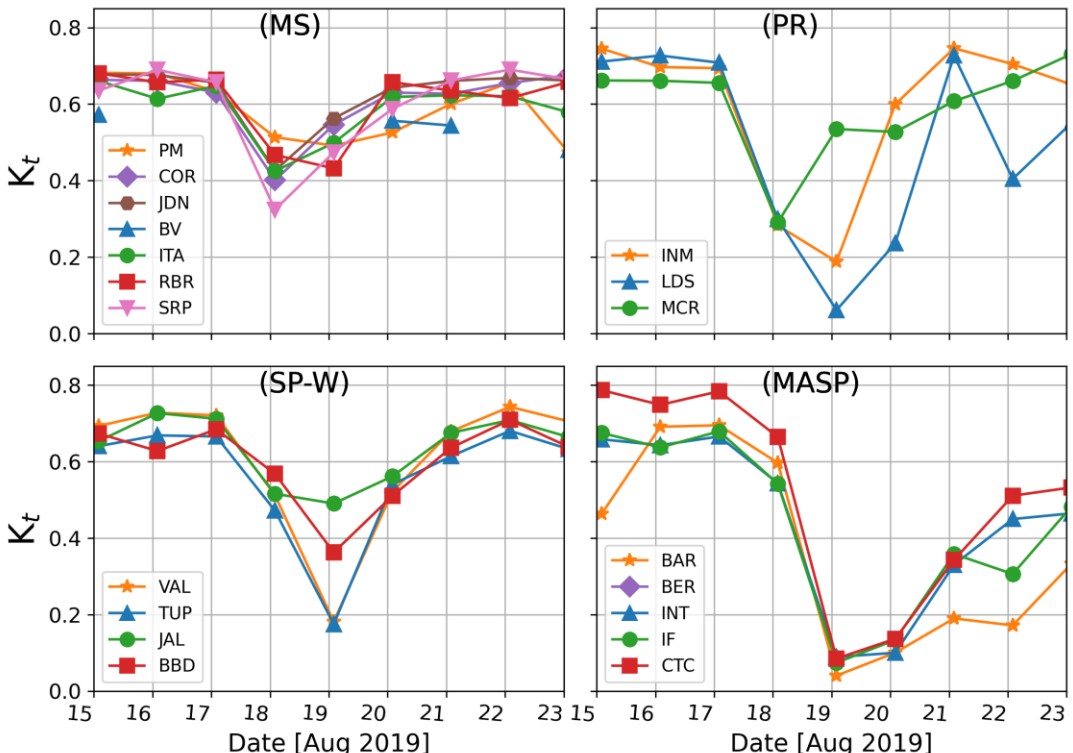

**Figure 9: Evolution of diurnal clearness index (K$_t$) in the 4 regions from August 15$^{th}$, 2019 to August 23$^{th}$,2019 computed from meteorological stations measurements network.**

## 3.3 Cloud radiative impacts and the darkness event on August 19 at MASP

The high-temporal-resolution (1-minute) measurements of global horizontal irradiance (GHI) at the CTC station are analysed and discussed in this section, with particular emphasis on their behaviour during the darkness event on August 19, 2019. This date is hereafter referred to as the "smoke-affected day".

In Figure 10, we present the 1-minute diurnal cycle of GHI, along with the average diurnal cycle for the 15 days before and after the smoke-affected day, all under all-sky conditions. Note that the darkness conditions (irradiances to zero) lasted around 40 minutes, starting near 15:00 LT. The average diurnal cycle of the preceding and following 15 days shows irradiances up to 150 W m$^{-2}$, after 15:00 LT. When the darkness event began, the cosine of solar zenith angle (CSZA) was 0.6, which corresponds to a moderate Sun elevation angle of around 36 degrees. The duration of the darkness event on August 19, 2019, was twice the maximum darkness period observed on any single day during the last two years of measurements.



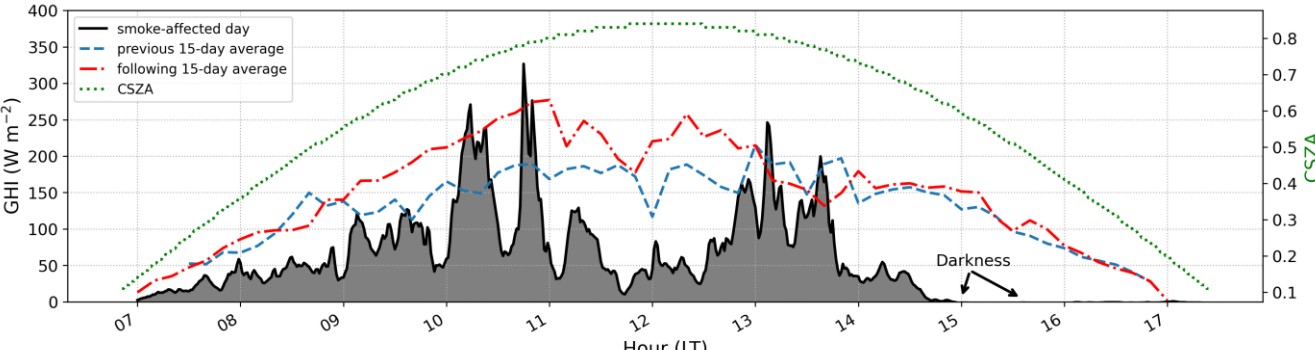


**Figure 10: Downward solar irradiance at the surface (GHI) during the smoke-affected day (August 19, 2019) and average of 15 days before and after for all sky conditions. In addition, the diurnal cycle of the cosine of the solar zenith angle (CSZA) for the smoke-affected day is shown.**

In Fig. 11, a comparison of 1-minute COD between overcast days and the smoke-affected day is shown. Overcast

days were selected as days with 100 % of cloud amount, including low clouds, throughout the entire day, confirmed by visual observations at the CTC station. A total of 6 overcast days with valid CNR4 measurements were found in the range 2018-2020.

Note, COD values rarely exceed 100 on the overcast days. However, on the smoke-affected day we observed exceptionally high COD values between 14:40 LT and 15:00 LT, with peaks reaching 300. After 15:00 LT, the irradiance

dropped to zero, making it impossible to retrieve COD data. This peak, near 300, was twice as high as the maximum COD observed on any of the other six overcast days. This significant increase in COD is clearly attributed to an extremely thick cloud band associated with the arrival of a cold front system in the MASP region. The passage of the cold front favoured the transport of smoke aerosols into the MASP region. Combined with convective activity, this likely enhanced the presence of optically thick clouds.


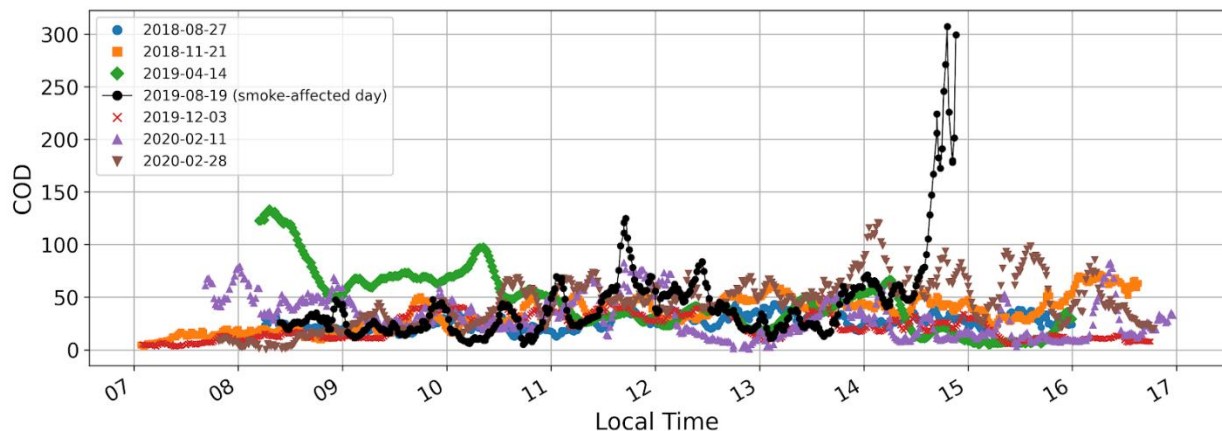

**Figure 11: Cloud optical depth (COD) retrieved from CNR4 measurements for the smoke-affected day (August 19, 2019) and 6 other overcast days found for the period of 2018-2020.**




The relationship of NCRE and COD was proposed by Mateos et al (2014) for estimating cloud radiative efficiency. The cloud
radiative efficiency quantifies the cloud radiative effect per unit of COD and can be used to analyse how the changes in aerosol
and cloud properties can change the cloud radiative properties. The radiative efficiency of clouds can be sensitive to CSZA,
aerosols, and cloud properties. For instance, we expect a more radiative effect per unit of COD at high CSZA values, as well
as in clouds with higher extinction properties (scattering and absorption). Errors in aerosol properties, estimated under clear-
sky conditions, lead to uncertainties in the COD retrieval and, consequently, in the NCRE.

Figure 12 shows the relationship between NCRE and COD for the overcast days selected, along with the aerosol properties
errors on the COD retrieval and the NCRE. The error analysis considered the propagation of uncertainties in aerosol optical
depth (AOD) and single scattering albedo (SSA). We assumed an uncertainty of 0.06 for AOD at 550 nm and 0.1 for SSA,
resulting in a 2% uncertainty in NCRE and a 15% in COD, respectively.

As expected, NCRE became more negative (indicating a stronger radiative effect) with increasing COD, until it remained
steady at a COD value above 100. The smoke-affected day exhibited the highest efficiency (NCRE per unit of COD). Note
that on this day, NCRE reached a value of -0.96, within the COD interval of 0-100 equivalent to a 96% reduction in downward
irradiance. By contrast, for the other overcast days within the same interval, NCRE reached -0.93. The differences in NCRE
between other overcast days and the smoke-affected day are approximately 7%, with statistical significance at the 5%
confidence level, as confirmed by the Mann-Whitney U test (Mann and Whitney, 1947). Note that the difference with the other
days is more evident for higher COD, where there is no overlap between the curves, including the error bars. To remove the
effect of the CSZA in the analysis we compared the smoke-affected day with an overcast day in the same season (August 27,
2018). Differences revealed a 6% of more cooling effect on the smoke-affected day, confirming that the high radiative
efficiency is due to changes in the radiative properties of clouds affected by the aerosol intrusion. The increased of extinction
capacity of shortwave radiation by clouds embedded in the biomass burning absorbing aerosol layers can explain the increase
of absorption and multiple scattering of cloud droplets. These results analysed here highlight the unique and extreme
characteristics of clouds embedded within smoke plumes and are also consistent with those observed in various regions.
Gautam et al. (2016) reported spectral modifications in clouds influenced by smoke, particularly in the visible (VIS) and
ultraviolet (UV) regions, with an enhancement of approximately 20%. Kaufman and Fraser (1997) a 28% increase in
reflectance in clouds affected by smoke in the Amazon and Cerrado basins of Brazil. Sarkar et al. (2022) found a 50–60%
increase in the fraction of low clouds over the Indian landmass due to elevated smoke levels. Twohy et al. (2021) documented
higher cloud droplet concentrations and smaller droplet sizes in low cumulus clouds during the wildfire season in the western
United States. Conrick et al. (2021) studied dense smoke over the Pacific Northwest and reported changes in cloud
microphysics, including an increase in small droplet concentrations, as well as thermodynamic effects that enhanced cloud
lifetimes.





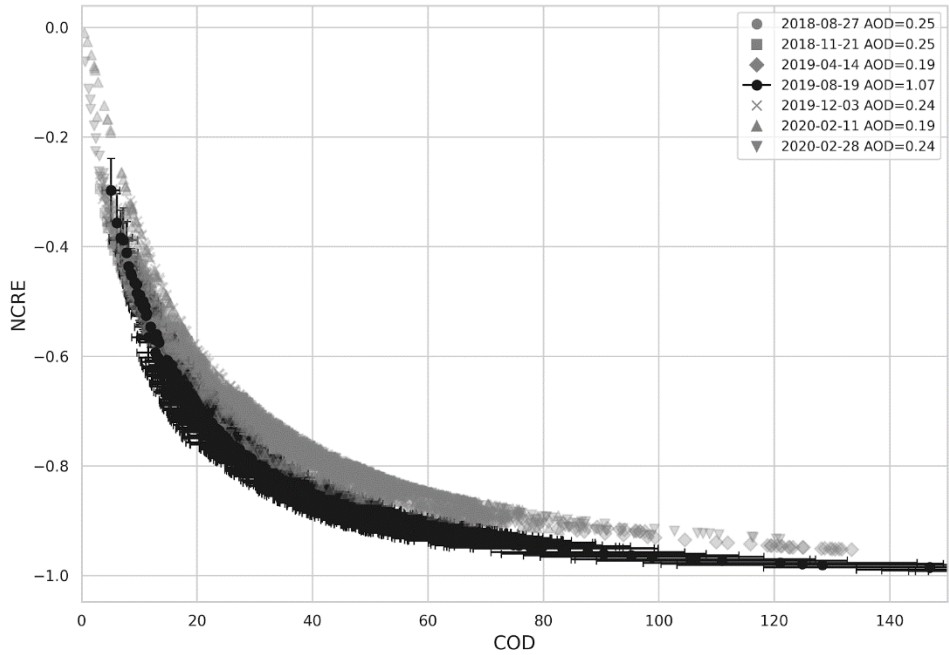

**Figure 12: Normalized cloud radiative effect (NCRE) versus cloud optical depth (COD) for smoke-affected day ('2019-08-19') and six other overcast days. Uncertainties associated with the estimation of each variable are also shown as vertical and horizontal bars.**

**Conclusions**

This study analyses a strong event of biomass burning aerosol transport from the central portions of South America during August 2019, with a particular focus on its intrusion into the southeastern of Brazil and the Metropolitan Area of São Paulo (MASP), where a 'black rain' and an extreme radiative darkness event occurred on August 19, 2019.

According to HYSPLIT back trajectories, air parcels arriving over MASP on this day followed a northwest-to-southeast trajectory, originating in the Brazilian Midwest, east of Bolivia, and north of Paraguay. The displacement of the smoke plume was tracked by ground based and satellite data.

The combination of clouds with smoke led to a strong radiative effect over MASP. The clearness index ($K_t$) dropped to 0.1, which is below the 1st percentile of the distribution for all-sky observations over the previous two years, indicating an extreme reduction in surface downward shortwave irradiance, particularly in MASP.

The combination of clouds and smoke also exhibited a 7% higher radiative extinction efficiency compared to other overcast days during the previous two years, suggesting that due to the presence of smoke particles, may have resulted in more effective solar radiation extinction per unit of cloud optical depth, particularly for higher COD values.

The August 19, 2019, darkness event in the MASP was a unique event when it understands of the complex interaction between smoke aerosol plumes and clouds. This study hopes to shed light on important aspects, from a radiative point of view, that can drive such types of impact. The lack of public policies and the recurrence of severe dry conditions in Brazil in recent years are again turning smoke transport towards MASP more frequent and intense, therefore, impacting for instance initiatives of







greenhouse gas mitigation using the sun as the main source of energy using photovoltaic systems. Further studies, especially

        in the field of modelling, are encouraged to evaluate the role of microphysical aspects on the discussed cloud smoke interaction

        and darkness event.

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

**Acknowledgements**

The authors would like to thank the AERONET, CALIPSO, and MODIS teams for providing valuable information on aerosol properties. We also thank the Instituto Nacional de Meteorologia (INMET) for granting access to the pyranometer
measurement network. Our appreciation extends to the staff at CIENTEC for their dedication in maintaining the deployed instruments. Special thanks are due to Dr. Bernard Mayer for the LibRadtran radiative transfer code.

This work has been supported by the Conselho Nacional de Desenvolvimento Científico e Tecnológico (CNPq) of the Brazilian government, under process numbers 140117/2019-9, 313005/2018-4, and 311984/2021-5, as well as by the Fundação de Amparo à Pesquisa do Estado de São Paulo (FAPESP), under process numbers 2018/16048-6, 2019/20794-8, 2016/18438-
0, and 2022/02609-1.



**Code/Data availability**

Data from MODIS and CERES can be downloaded from NASA's Langley Research Center (LARC) at https://ceres.larc.nasa.gov/data/.

Surface irradiance data can be accessed at https://bdmep.inmet.gov.br.

CALIPSO data is available at https://www.eorc.jaxa.jp/EARTHCARE/A-train/A-train_monitor_e.html.

Aerosol Optical Depth (AOD) data from MERRA2 can be accessed at https://goldsmr4.gesdisc.eosdis.nasa.gov/data/MERRA2/M2T1NXAER.5.12.4.

GOES-16 data is available on Google Cloud Storage at gs://gcp-public-data-goes-16/ABI-L2-MCMIPF.

ERA5 reanalysis data can be obtained from the Climate Data Store at https://cds.climate.copernicus.eu.

The radiative transfer code LibRadtran can be downloaded at http://www.libradtran.org/doku.php?id=download.

**Author contribution**

J R conceptualized the study, conducted the computations of the radiative transfer processes, and drafted the initial manuscript.

G L focused on the HYSPLIT analysis, the analysis of data from CERES and MODIS, and the identification of the aerosol smoke plume source. M A contributed to the development of the methodology, verified and corrected computations, and provided significant support in drafting the manuscript. N E helped in the theoretical analysis of the radiative processes and contributed extensively to writing and presenting the results.

**Competing interests**

The authors declare no competing interests