# Peer review of "Biomass burning smoke transport and radiative impact over the city of Sao Paulo: An extreme event case study"

_EGUsphere, 2025_

## Referee Comment (RC2)

**General comments**

The paper entitled "Biomass burning smoke transport and radiative impact over the city of Sao Paulo: An extreme event case study" (EGUsphere preprint, doi:10.5194/egusphere-2025-9) addresses an extreme and highly relevant real-world event (São Paulo's "dark day" in August 2019), providing valuable insights into biomass burning smoke transport and its radiative effects. The overall study is regionally important and broadly relevant for atmospheric sciences, climate impact studies, and urban planning. The work is overall sound, and I have some points to raise. The results are of interest to the scientific community and i would suggest publication, however following minor revisions.

**Specific comments & Technical corrections:**

I recommend paying attention to the Figures captions, providing clear sentences that highlight what the Figures describe in a clear and understandable way for the reader.

More discussion relative to the satellite data use is needed:

- Throughout the manuscript, the satellite CALIPSO is referenced correctly as the platform providing vertical aerosol profiling data. However, the manuscript does not mention the instrument name CALIOP (Cloud-Aerosol Lidar with Orthogonal Polarization), which is the actual lidar sensor onboard CALIPSO responsible for the backscatter measurements. Since the data product used (e.g., total attenuated backscatter at 532 nm) originates from CALIOP, it is scientifically accurate and standard in atmospheric literature to reference both the platform (CALIPSO) and the instrument (CALIOP) where appropriate. This addition would ensure clarity for readers, especially those less familiar with satellite instruments.

  Please replace "Winker et al., 2009" instead of "Winker et al., 2007" and add in the reference section the appropriate citation:
  - *"Winker, D. M., Vaughan, M. A., Omar, A., Hu, Y., Powell, K. A., Liu, Z., Hunt, W. H., and Young, S. A.: Overview of the CALIPSO Mission and CALIOP Data Processing Algorithms, J. Atmos. Ocean. Tech., 26, 2310–2323, https://doi.org/10.1175/2009JTECHA1281.1, 2009."*

- A short discussion regarding the quality flags and thresholds applied in the datasets (e.g. CALIOP profiling, MODIS AOD) is missing. Be more specific and add some relevant text within the manuscript. Additionally, it is not mentioned anywhere in the text how the data is used to categorize the detected layer as

smoke (based on depolarization? Lidar Ratio?). It's strongly reccomended to include a Level-2 CALIOP ALAy/Apro data analysis highlighting the event.

- The Dark Target (DT) algorithm is applicable for the retrieval of aerosol loading and properties over dark surfaces, including ocean water and vegetated land. Have you taken the necessary limitations into account when using the products?

- What is the role of using FRP products? Only for thermal hot spots identification? Please specify. You can use FRP to provide fire intensity and correlate with fire activity.

**Code/Data availability section:** For each of the data used in the work, please provide the specific collection and sources thereof along with the corresponding DOI as well as the date of last access.

**Table 2:** Recommend to modify the header as: "Station name, "Latitude (°)", "Longitude (°)" and "Region acronym".

**Table 3:** I propose a short Table modification. i) $1^{st}$ column: Keep a specific format style about the dataset's description, also add the related collection/version and provide the appropriate DOI for each of the satellite products used within the study, ii) $2^{nd}$ Column: You refer only to "spatial resolution", keep this as column label, iii) $3^{rd}$ column: Keep a specific format style about the source's description and iv) $4^{th}$ column: header column label should start with a capital letter, check for typo , comma, parentheses etc

**Figure 2:**

- Add a title line adding the Time (in UTC) and overpass mode (Nightime or daytime) in the bottom Figure corresponding to CALIOP Total attenuated signal.
- Rephrase "… the rectangle highlights…" to "…the dashed rectangle…"
- "…. with the aerosol layer region highlighted": Be clearer, add the  color (select an appropriate visible color) of the "highlight" and use the same color as rectangle box in the Quicklook attenuated signals figure (bottom) where the smoke plume is detected.
- "In the bottom panel, arrows indicate the position of the aerosol smoke plume, as classified by the CALIPSO algorithm": This figure does not indicate the presence of smoke, we do not get this information clearly with this figure. It is the total attenuated backscatter signal that presents the recovered signals as seen by the CALIOP lidar and does not distinguish

and classify into aerosol types. I would suggest using the figure referred to as "aerosol subtype mask" and it is clear to the reader the classification of the detected layer as smoke, briefly citing the CALIOP capability within the text (also provide the appr. Reference), or keep both graphs together.

**Figure 3:** I recommend increasing the font size for the colorbar captions as well as within the graph so that they are clearly visible to the reader.

**Figure 4:** For each individual map within the Figure, place captions (a-c) for a comprehensive description. Also add here that you refer to a.g.l.

**Figure 5:** For each individual map within the Figure, place captions (a-d) for a comprehensive description.

**L29:** Modify "Aerosols from BB..." to " Aerosols emitted from BB sources..."

**Line 89:** What is refer for "Version 2.0"? I guess you mean Level. 2?

**Line 106:** Rephrase the sentence to: "Table 3 provides an overview of the satellite products and reanalysis datasets utilized in this study." And ".....were averaged, applying distance-based weighting to better capture local atmospheric conditions."

**Line 115:** add the last access for the mentioned web-link data source.

**Line 119:** "..at specific local times..". add the overpass time respectively and also the swath.

**Line 120:** Please refer to the products exactly as in the table regarding AOD.

**Line 120:** Rephrase "..using the dark target methodology.." to "based on Dark Target (DT) algorithm.."

**Line 126:** "..version 4-10.." . There are also more updated versions (e.g. 4.51). Why do you choose 4.10? Please clarify.

**Line 128:** What is the role of using FRP products? Only for thermal hot spots identification? Please specify.

**Line 202:** "...the fire spots". You have mentioned previously in the manuscript that you also use the FRP (Fire Radiative Product) obtained by MODIS. Why is this information not used instead of simply indicating the presence of "fire hot spots"? You can also add another colorbar to the graph with the fire intensity (in MW), this would add one more information to the study regarding the severity and intensity of the fires. If this is not in your intentions, please justify your reference to the use of the FRP product.

**Line 203:** Does the CALIPSO information refer to h.a.s.l or a.g.l? please, clarify.

**Line 205:** Also mention that these heights correspond to a.g.l.